# Estimation of the Chelating Ability of an Extract from *Aronia melanocarpa* L. Berries and Its Main Polyphenolic Ingredients Towards Ions of Zinc and Copper

**DOI:** 10.3390/molecules25071507

**Published:** 2020-03-26

**Authors:** Sylwia Borowska, Michał Tomczyk, Jakub W. Strawa, Małgorzata M. Brzóska

**Affiliations:** 1Department of Toxicology, Medical University of Bialystok, Adama Mickiewicza 2C, 15-222 Bialystok, Poland; sylwiaborowska86@tlen.pl; 2Department of Pharmacognosy, Medical University of Bialystok, Adama Mickiewicza 2A, 15-230 Bialystok, Poland; jakub.strawa@umb.edu.pl (J.W.S.); michal.tomczyk@umb.edu.pl (M.T.)

**Keywords:** *Aronia melanocarpa* berries extract, chelating ability, complexation, polyphenols, cyanidin 3-*O*-*β*-galactoside, quercetin, copper, zinc

## Abstract

Previously, we have revealed that prolonged administration of a polyphenol-rich 0.1% extract from the berries of *Aronia melanocarpa* L. (chokeberries) alone and under chronic exposure to cadmium influences the body status of zinc (Zn) and copper (Cu). The aim of this study was to evaluate, in an in vitro model, the chelating properties of the extract (0.05% and 0.1%) and its main polyphenolic ingredients (cyanidin 3-*O*-*β*-galactoside, chlorogenic acid, neochlorogenic acid, (+)-catechin, (−)-epicatechin, quercetin, and kaempferol) regarding divalent ions of Zn (Zn^2+^) and Cu (Cu^2+^) at pH reflecting physiological conditions at the gastrointestinal tract such as 2 (empty stomach), 5.5 (full stomach), and 8 (duodenum). The study has revealed that the extract from *Aronia* berries, as well as cyanidin 3-*O*-*β*-galactoside and quercetin, can bind Zn^2+^ and Cu^2+^, but only at pH 5.5. Moreover, kaempferol was able to chelate Zn^2+^ at pH 5.5; however, this ability was weaker than those of cyanidin 3-*O*-*β*-galactoside and quercetin. The ability of the chokeberry extract to chelate Zn^2+^ and Cu^2+^ may be explained, at least partially, by the presence of polyphenols such as anthocyanin derivatives of cyanidin and quercetin. The findings seem to suggest that *Aronia* products, used as supplements of a diet, should be consumed before meals, and particular attention should be paid to adequate intake of Zn and Cu under prolonged consumption of these products to avoid deficiency of both bioelements in the body due to their complexation by chokeberry ingredients in the lumen of the gastrointestinal tract.

## 1. Introduction

Nowadays, a growing interest in the possibility of joining biologically active compounds, present in medicinal plants, into a daily diet to prevent from the development of civilization diseases and to support their treatment has been observed [1,2,3,4]. One of the most important groups of such compounds are polyphenols. These compounds, mainly due to their antioxidative and anti-inflammatory properties, are effective in the protection against disorders in the cardiovascular, digestive, nervous, and reproductive systems, as well as against hypertension, cirrhosis, overweight and obesity, diabetes, skin troubles, and cancers [1,2,3,5]. Moreover, increasing attention has been focused on polyphenol-rich products as possible effective factors in the protection against action of some xenobiotics, including common pollutants of the natural environment and dietary products such as toxic heavy metals [1,4,6,7,8].

It seems reasonable that, among plants abundant in polyphenols, special attention should be paid to *Aronia melanocarpa* L. (Rosaceae), the berries of which (chokeberries) are one of the richest sources of these compounds [1,2,4,9]. Chokeberries have proven beneficial impact on health and thus they are widely recommended as functional food [1,2,3,5]. Results of experimental studies indicate that *Aronia* products seem to be very promising agents in the protection from unfavorable effects of exposure to heavy metals, including cadmium (Cd) [1,4,6,8,10,11,12,13,14,15], belonging to the main environmental pollutants in industrialized countries [7]. In a study conducted in a rat model of low-level and moderate lifetime human exposure to Cd (1 and 5 mg Cd/kg diet, respectively, for up to 24 months) we have revealed that the administration of a 0.1% extract from the berries of *A. melanocarpa* (AE) during the exposure to Cd decreased the absorption of this xenobiotic from the gastrointestinal tract and its accumulation in the body [10], protected from damage to the bone [11,12] and liver [14,15], as well as completely or partially prevented most of the changes in the metabolism of zinc (Zn) and copper (Cu) [13].

Based on the available data, it can be supposed that the protective action of polyphenols and polyphenol-rich products against heavy-metal toxicity is mainly connected with chelating abilities and antioxidative properties of these compounds [1,4,6,8,12,14,15]. Polyphenols, due to the presence of hydroxyl (-OH) groups, are able to form complexes with ions of toxic metals, including Cd ions (Cd^2+^), as well as with ions of divalent bioelements, such as Zn (Zn^2+^) and Cu (Cu^2+^) [6,16,17]. Complexation of metals by these compounds may result in a decrease in the absorption from the gastrointestinal tract of the former, an increase in their elimination with urine, and a decrease in the concentrations of biologically available Zn^2+^ and Cu^2+^ in the extra- and intracellular fluids [6]. The complexation of metal ions may prevent from accumulation and unfavorable action of toxic metals; however, at the same time, it may unfortunately lead to bioelement deficiency in the organism [6,13]. Some, but limited thus far, data show that repeated consumption of *Aronia* products may negatively influence the body status of Zn and Cu [5,6]. Kowalczyk and coauthors [5] have reported that the daily consumption of 240 mg of *Aronia* anthocyanins for 30 days increased Zn concentration and decreased Cu concentration in the red blood cells in men with hypercholesterolemia. We have revealed that the prolonged (3–24 months) administration of the AE alone generally had no influence on the body status of Zn and Cu. However, the extract intake temporarily decreased the bioavailability of both bioelements and influenced (decreased or increased) their concentrations in some tissues and biological fluids of rats (Appendix A) [13]. Moreover, as mentioned above, the administration of the 0.1% AE under the low-level and moderate exposure to Cd completely or partially protected against unfavorable impact of this toxic metal on the metabolism of Zn and Cu, but it also influenced (increased or decreased) some of the indices of the metabolism of these bioelements unchanged by this xenobiotic alone [13]. Among others, the 24-month intake of the extract under the moderate exposure to Cd decreased (by 10%) the total content of Cu in internal organs [13]. Based on our findings [13] and properties of ingredients of the AE [1,6,8], we have hypothesized that the chokeberry extract may influence the body status of Zn and Cu, including their absorption from the digestive tract, due to its ability to bind these bioelements resulting from its abundance in polyphenols.

Taking into account the above findings on the impact of *Aronia* products on the metabolism of Zn and Cu [5,13] together with the widely recommended consumption of these products in order to protect or improve the health status [1,2], it seems very important to recognize the possible interactions between ingredients of chokeberries and these bioelements. It is also essential due to the fact that Zn and Cu deficiencies still occur in a diet of the general population [18,19]. Thus, the aim of the present study was to investigate whether the 0.1% AE may form complexes with Zn^2+^ and Cu^2+^. Because, in the case of oral intake, the lumen of the gastrointestinal tract is theoretically the first place of interactions between ingredients of the AE and Zn^2+^ or Cu^2+^, the chelating ability of the extract was evaluated at pH reflecting the conditions noticed in different parts of the digestive system (empty or full stomach and duodenum). Due to the fact that the intake of *Aronia* products, Zn, and Cu with a diet may vary, the experiment was performed with the use of different concentrations of the AE and both bioelements in order to establish whether the chelating ability of the extract is dependent on these factors. Moreover, to explain which of the polyphenolic compounds present in the AE may form complexes with Zn^2+^ and Cu^2+^, the chelating abilities of the main polyphenols present in the extract towards ions of both bioelements were estimated. According to our knowledge, a similar study has not been conducted until now.

## 2. Results

### 2.1. Zn^2+^ Chelation by AE and Polyphenolic Compounds Present in the Extract

It is evident from the data presented in Figure 1 and Figure 2 and Table 1 and Appendix A that the AE (0.05% and 0.1%), cyanidin 3-*O-β*-galactoside (C3G), quercetin (Q), and kaempferol (K) were able to form complexes with Zn^2+^ at the concentrations of 0.01, 0.1, and 3 mM at pH 5.5, but not at pH 2 and 8. The length of the maximum absorption of the AE without Zn^2+^ (283 and 527 nm) and with Zn^2+^ (283 and 579 nm) corresponds well with the maximum absorption of C3G without Zn^2+^ (282 and 528 nm) and with Zn^2+^ (281 and 585 nm; Figure 1 and Figure 2, Appendix A). Chlorogenic acid (CA), neochlorogenic acid (NCA), (-)-epicatechin, and (+)-catechin did not chelate Zn^2+^ at pH 2, 5.5, and 8 at all studied concentrations (Figure 2, Table 1 and Appendix A). Ethylenediaminetetraacetic acid (EDTA) formed complexes with Zn^2+^ at all used concentrations (Figure 2, Table 1, Appendix A).

### 2.2. Cu^2+^ Chelation by AE and Polyphenolic Compounds Present in the Extract

The AE (0.05% and 0.1%), C3G, and Q formed complexes with Cu^2+^ at the concentrations of 0.01, 0.05, and 0.5 mM at pH 5.5 but not at pH 2 and 8 (Figure 3 and Figure 4, Table 1 and Appendix A). The length of the maximum absorption of the AE without Cu^2+^ (283 and 527 nm) and with Cu^2+^ (283 and 583 nm) corresponds well with the maximum absorption of C3G without Cu^2+^ (282 and 528 nm) and with Cu^2+^ at pH 5.5 (265 and 585 nm; Figure 3 and Figure 4, Appendix A). CA, NCA, (−)-epicatechin, (+)-catechin, and K did not chelate Cu^2+^ at pH 2, 5.5, and 8 at all studied concentrations (Figure 4, Table 1 and Appendix A). EDTA formed complexes with Cu^2+^ at all used concentrations (Figure 4, Table 1, Appendix A).

## 3. Discussion

The most important achievement of the present study is the finding that the 0.1% AE was able to form complexes with Zn^2+^ and Cu^2+^ at pH 5.5, which seems to suggest that the extract may chelate these bioelements in the lumen of the gastrointestinal tract. The present study also provides an explanation as to which of the *Aronia* polyphenols may be responsible for the extract chelating ability and which of them do not have such properties. A detailed analysis of the results, taking into account the chemical structure of the used polyphenols, allows for the conclusion that the ability of the 0.1% AE to chelate Zn^2+^ and Cu^2+^ may be explained, at least partially, by the presence of anthocyanin derivatives of cyanidin and Q.

Because the aim of this study was to investigate whether the 0.1% AE may form complexes with Zn^2+^ and Cu^2+^, in order to explain the results of our earlier research concerning the influence of prolonged intake of the extract on the body status of Zn and Cu under low-level and moderate chronic exposure to Cd and without treatment with this toxic heavy metal [13], we have created an in vitro model reflecting the conditions of the gastrointestinal tract. That is why the ability of the AE to form complexes with Zn^2+^ and Cu^2+^ was evaluated at pH 2, 5.5, and 8, reflecting the conditions noticed in various parts of the digestive system in different situations (empty stomach—pH 1–2; full stomach—pH 5.5; duodenum—pH 6–8) [20,21] and the concentrations of 0.01, 0.1, and 3 mM Zn^2+^ and 0.01, 0.05, and 0.5 mM Cu^2+^ were used. Because Zn and Cu are noticed in a meal at the levels of 3–25 and 0.5–15 mM, respectively [22,23,24], these values may be the highest concentrations theoretically present in the upper parts of the lumen of the digestive tract. The concentration of 0.01 mM Zn has been detected in human gastric juice [25], while the concentration of 0.1 mM Zn was noted in the human prostate [26]. The concentration of 0.01 mM Cu corresponds to its concentration in human blood [27], while the concentration of 0.05 mM of this element was chosen as an intermediate value between that detected in diet and that determined in blood.

The use of two different concentrations of the AE (0.05% and 0.1%) in the first stage of the study allowed evaluation of the impact of the extract concentration on its chelating properties towards Zn^2+^ and Cu^2+^. Because a lack of differences between the complexing abilities of the 0.1% and 0.05% AE towards Zn^2+^ and Cu^2+^ has been revealed, the abilities of the main *Aronia* polyphenols to complex these ions were evaluated in the second stage only at the concentrations corresponding to their concentrations in the 0.1% AE. The use of NCA and CA, isomers belonging to the group of hydroxycinnamic acids [2,28], at the same concentrations allowed for comparison of chelating abilities of these two compounds. Because proanthocyanidins present in chokeberries constitute oligomers built from monomers that are derivatives of (−)-epicatechin and (+)-catechin [1], both (−)-epicatechin and (+)-catechin were used at the highest concentration (0.013%) in which they could theoretically be found in the 0.1% AE [11]. The applied concentration of Q (0.002%), a flavonol derivative with a dihydroxyl group in the B ring, is equivalent to the concentration of flavonol derivatives identified in the 0.1% AE (Table 2) [11]. The use of K (a flavonol derivative with the same structure as Q, but with only a single -OH group in the B ring) at the same concentration as Q (possessing a dihydroxyl group in the B ring), i.e., 0.002%, allowed to explain whether the -OH group in the B ring may influence chelating properties of polyphenols.

The ability of anthocyanin derivatives of cyanidin (C3G) and Q to complex Zn^2+^ and Cu^2+^ (at pH 5.5) is connected with the presence of ortho-dihydroxyl groups in the B ring in structure of these compounds (Figure 5) [6]. Our finding that both Q and cyanidin derivatives may form complexes with Zn^2+^ or Cu^2+^ are in accordance with the results of other authors [16,17]. It has been recognized that C3G may form complexes with ions of divalent metals, probably due to the presence of ortho-dihydroxyl groups in the B ring [6]. The inability revealed in the present study of NCA to complex Zn^2+^ and Cu^2+^ and of K to bind Cu^2+^, as they are polyphenols without an ortho-dihydroxyl group in the B ring, confirms the crucial role of this group in the B ring in the process of metal complexation (Figure 5). However, the ability of K (possessing only a single -OH group in the B ring) to bind Zn^2+^ at pH 5.5 shows that the presence of the ortho-dihydroxyl group in the B ring is not absolutely necessary. It should be clearly underlined that detailed analysis of the received UV-Vis absorption spectra (Figure 2, Appendix A) seems to indicate that C3G and Q, possessing this structure, are stronger chelators of Zn^2+^ than K is. Moreover, the fact that CA was unable to bind Zn^2+^ and Cu^2+^ in all studied pH seems to indicate that the presence of a dihydroxyl group in the polyphenolic structure does not guarantee its chelating properties. According to our knowledge, there are no data in scientific literature concerning the chelating properties of CA towards bioelements. Thus, we are unable to explain why this compound was unable to complex divalent ions of Zn and Cu.

The finding that the length of the maximum absorption of the 0.05% and 0.1% AE without Zn^2+^ or Cu^2+^ and with Zn^2+^ or Cu^2+^ corresponded well with the maximum absorption of C3G without and with these ions suggests that cyanidin derivatives, belonging to anthocyanins (which constitute one of the main groups of polyphenolic compounds present in the AE (Table 2; [11]), may be responsible for the chelating properties of the extract towards Zn^2+^ and Cu^2+^. Taking into account the fact that the length of the maximum absorption of Q did not correspond well with the length of the maximum absorption of the 0.1% AE and that this polyphenolic compound is present at low concentration in the extract (total flavonoids: 21.94 ± 0.98 μg/L; Table 2; [11]), this polyphenol may be responsible to a lesser extent than cyanidin derivatives for the chelating properties of the extract towards Zn^2+^ and Cu^2+^. However, it is K that seems to be the weakest chelator of Zn^2+^. The lack of binding of Cu^2+^ by K might result from weak complexing capacity of this compound towards divalent ions of metals. It is important to underline that the noted lack of differences in the ability of binding Zn^2+^ and Cu^2+^ dependent on the used concentrations of these bioelements shows that both AE and its chosen polyphenolic ingredients (C3G, Q, and K in the case of Zn^2+^; C3G and Q in the case of Cu^2+^) are able to complex these elements within a relatively wide range of concentrations.

The results of this study are useful for the explanation of the results of our earlier research concerning the influence of prolonged intake of the 0.1% AE on the body status of Zn and Cu, as well as the impact of this extract on the metabolism of these bioelements under low-level and moderate chronic exposure to Cd [13]. The transitional decrease in the bioavailability (declined apparent absorption and retention in the body), increase in the urinary excretion and decrease in the concentrations of Zn and/or Cu in some tissues and biological fluids of rats administered with the AE alone (Appendix A) [11] may be explained by formation of complexes between the extract ingredients and these bioelements in the body, including mainly the lumen of the digestive tract.

As previously mentioned, the administration of the 0.1% AE completely or partially prevented most of the changes in the metabolism of Zn and Cu caused by Cd in rats; however, the AE did not always have a protective effect against Cd-induced changes in the body status of these elements, and sometimes it also influenced some of the indices of their metabolism unchanged by Cd alone [13]. These may be connected with the complexation of Zn and Cu by ingredients of the extract, including cyanidin derivatives and Q, as well as K in the case of Zn. Our previous finding of decreased total content of Cu in internal organs due to the administration of the AE under moderate exposure to Cd for 24 months compared not only with the group exposed to Cd but also with the control group [13], together with the ability revealed in the present study of the extract to bind Cu^2+^, allows to conclude that prolonged consumption of *Aronia* products under exposure to this xenobiotic may lead to Cu deficiency in the organism.

The fact that complexation of Zn^2+^ and Cu^2+^ occurs at pH 5.5 indicates that complexes between polyphenolic compounds present in the AE and these bioelements may be formed in the stomach after a meal (when a pH value grows from strong acidic up to 5.5). This finding may suggest that the AE should not be consumed during a meal. Moreover, the complexation of Zn^2+^ and Cu^2+^ may take place not only in the lumen of the digestive tract but probably also in the internal organs and tissues [13,25]. The results of our study seem to indicate that the chelation of both bioelements by polyphenolic compounds present in the extract may occur at the concentrations in which Zn and Cu are present in the human body (gastric juice, blood, and prostate).

The knowledge on the chelating properties of the 0.1% AE in different pH may be useful also for the prediction of possible interactions between the extract or its chosen ingredients and Zn^2+^ or Cu^2+^ under not only physiological but also pathological conditions, due to the fact that various disorders may change pH in the digestive tract [20]. Despite the fact that, in the present study, the complexation of Zn^2+^ and Cu^2+^ by the extract has not been observed at pH reflecting conditions specific for duodenum, it cannot be excluded that the chelation of these ions by the 0.1% AE may take place in the duodenum under the specific conditions when the pH value in this part of the gastrointestinal tract approaches the value of 5.5. This may occur in patients with chronic pancreatitis, who have exhibited pH values of 5–6 in this part of the gastrointestinal tract [20].

Apart from polyphenols, other bioactive compounds like fibers and pectins may also be responsible for chelating properties of chokeberries [29]. It cannot be excluded that other components of *Aronia* berries such as dietary fiber, tannins, and pectins can also bind Zn^2+^ and Cu^2+^, decreasing their bioavailability. It has been revealed that pomace from blackcurrants rich in dietary fiber is able to bind Cu^2+^ at 69% and Zn^2+^ at 21% at pH 2.0 (what was evaluated as a sorption—a ratio between the metal content in a sample and in an adsorbate demonstrated as a percentage value) [30]. Moreover, it has been reported that the ability of fiber to bind Cu^2+^ decreased significantly (a drop in sorption from 69% to 8%) with a decline in pH value (from 2 to 1) [30]. However, better binding properties of dietary fibers (from potato, rye bran, wheat bran, rice bran, corn bran, soybeans, and oat hulls) towards Zn^2+^ and Cu^2+^ have generally been observed at higher pH (6.8 and 4.5) than at lower pH (0.65, 2.2, and 3.2) [31,32], which corresponds with our results concerning chelating properties of the 0.1% AE and polyphenols present in this extract, which were able to complex Zn^2+^ and Cu^2+^ at higher pH (5.5) but not at lower pH (2.0).

Wider discussion of the results of this study is presently impossible due to a lack of data on chelating properties of other plant extracts rich in polyphenolic compounds (e.g., extracts from black elderberry, blackcurrant, blackberry, blueberry, and raspberry) towards Zn^2+^ and Cu^2+^, especially in living organisms. The evaluation of interactions between plant extracts rich in polyphenols and these bioelements is particularly important at pH reflecting the environment of the gastrointestinal tract, since plant products are joined to a daily diet which may result in a decrease in trace element bioavailability from a meal. Our findings may suggest that *Aronia* products (not only extracts, but also juices) should be consumed on an empty stomach and not by subjects suffering from chronic pancreatitis with lower than normal pH value in duodenum. Anthocyanins—derivatives of cyanidin—are also present at high concentrations in other popular foodstuffs (blackcurrant pomace, red wine, black raspberry, cranberry, blueberry, bilberry, lingonberry, red onion, Mexican oregano, as well as cocoa powder and black chocolate) [3], so it might be expected that products made from them may also form complexes with Zn^2+^ and Cu^2+^. This is particularly important due to the fact that a shortage of Zn and/or Cu in a diet is still a current problem concerning the general population [18,19].

The knowledge about interactions between AE, as well as other food products rich in cyanidin derivatives, Q, and K and Zn^2+^ or Cu^2+^ in the digestive tract may be useful in planning a proper daily diet, and it may allow for establishing a proper menu that reduces the risk of negative consequences which may result from mixing of some food products. Presently, there is a lack of data on this issue obtained from human subjects. The results of the present study may suggest that the prolonged consumption of AE may lead to Zn and Cu deficiency as a consequence of their chelation by the extract. However, it is also very important to underline that, in the case of a diet containing excess amounts of Zn or Cu, complexation of these elements by ingredients of AE may provide protection against their excessive gastrointestinal absorption and unfavorable action in the body.

The findings of our research have great scientific and practical value. They seem to show that it is very important whether polyphenol-rich products, including *Aronia* extracts, currently broadly advised by nutritionists and clinicians as a functional food in the prevention and treatment of civilization diseases [1,2], are consumed before, during, or after a meal. The results may suggest that products rich in polyphenols, including extracts and juices, as a part of a daily diet or supplements, should be consumed on an empty stomach. Owing to the fact that deficiencies of bioelements in a diet still occur in the general population [18,19], this issue is particularly important in the case of consumption of polyphenols by subjects suffering from bioelement deficiency. The findings of the present study suggest that proper amounts of microelements such as Zn and Cu should be delivered into the organism during the consumption of products rich in polyphenols to prevent from their deficiency or from intensification of existing shortage of these bioelements. So far, the complexation of bioelements by polyphenols in the lumen of the gastrointestinal tract has not been studied in the living organism, but our findings show the necessity of such study in humans.

We are aware not only of novelty and achievements of our study, but also of its limitations. The main limitation of our study is the fact that the potential chelating properties of the AE and particular polyphenolic compounds towards Zn^2+^ and Cu^2+^ were investigated only with the use of one method, and they have not been estimated and confirmed by other methods. Moreover, the chemical nature (e.g., stability, ligand dissociation constants) of the formed complexes has not been assessed. The fact that we are unable to explain the lack of complexation of Zn^2+^ and Cu^2+^ by CA possessing a dihydroxyl group is also a limitation. Moreover, we were unable to explain why (−)-epicatechin and (+)-catechin possessing a free 3′,4′-dihydroxyl group in the B ring had no chelating properties towards both bioelements. This may result from the fact that proanthocyanidins in chokeberries exist in the form of oligomers that are derivatives of (−)-epicatechin and (+)-catechin [33], but in the present study we used monomers of (−)-epicatechin and (+)-catechin. We are aware that some of our considerations carried out in the discussion of the results of the present study are in the sphere of suppositions that require scientific confirmation; however, they indicate the possible health implications of bioelement complexation.

In conclusion, the study not only confirmed our hypothesis regarding the possibility of binding Zn^2+^ and Cu^2+^ by the extract from *A. melanocarpa* berries, but also allowed to clarify which of the polyphenolic compounds present in it can form complexes with ions of these bioelements and under what pH conditions they do so. The extract can complex Zn^2+^ and Cu^2+^ at pH 5.5, reflecting conditions noticed in a full stomach, and this ability may be explained, at least partially, by the presence of polyphenols such as anthocyanin derivatives of cyanidin and quercetin. The findings of the present study seem to suggest that *Aronia* products, used as supplements of a diet, should be consumed on an empty stomach, and particular attention should be paid to adequate intake of Zn and Cu under prolonged consumption of these products to avoid deficiency in both bioelements in the body due to their complexation by chokeberry ingredients in the lumen of the gastrointestinal tract. This may especially apply to people at risk of Zn and Cu deficiency. Moreover, based on the results, it can be concluded that consumption of *Aronia* products or another products rich in cyanidin derivatives and Q in the case of excess consumption of Zn or Cu may provide protection against their excessive gastrointestinal absorption and unfavorable action in the body due to complexation of these elements. However, further studies, including examination of the chemical nature of complexation, are needed to recognize the health implications of the complexing ability of *Aronia* extract and its ingredients towards Zn and Cu.

## 4. Materials and Methods

### 4.1. Chemicals

Lyophilized extract from the berries of *A. melanocarpa* (Rosaceae) (Adamed Consumer Healthcare, Tuszyn, Poland; Certificate KJ 4/2010, Batch No. M100703) and pure polyphenolic compounds (Sigma-Aldrich, St. Louis, MO, USA) such as C3G, CA (3-(3,4-dihydroxycinnamoyl)-quinic acid), NCA (5-O-(trans-3,4-dihydroxycinnamoyl)-D-quinic acid), (+)-catechin ((+)-trans-3,3′,4′,5,7-pentahydroxyflavane), (−)-epicatechin ((−)-cis-3,3′,4′,5,7-pentahydroxyflavane), Q (2-(3,4-dihydroxyphenyl)-3,5,7-trihydroxy-4H-1-benzopyran-4-one, 3,3′,4′,5,6-pentahydroxy-flavone), and K (3,4′,5,7-tetrahydroxyflavone, 3,5,7-trihydroxy-2-(4-hydroxyphenyl)-4H-1-benzopyran-4-one), as well as pure EDTA (Avantor Performance Materials Poland S.A., Gliwice, Poland) were used. Zn and Cu salts such as anhydrous zinc chloride (ZnCl_2_) and anhydrous copper(II) sulfate(VI) (CuSO_4_) were purchased from Avantor Performance Materials Poland S.A (Gliwice, Poland). Hydrochloric acid (HCl) and sodium hydroxide (NaOH) were provided by Sigma-Aldrich (St. Louis, MO, USA). As a solvent of the AE, pure polyphenolic compounds, EDTA, and salts of bioelements, a solution of methanol (Avantor Performance Materials Poland S.A., Gliwice, Poland) and ultra-pure water (H_2_O; MAXIMA purification system; ELGA, Lane End, United Kingdom) mixed at the ratio of 7:3 was used. Buffer solutions of pH 4.01 and 10.01 were provided by Mettler Toledo (Schwarzenbach, Switzerland).

### 4.2. Ingredients of AE

Polyphenolic profile of the 0.1% AE used in this study is presented in Table 2 [11]. The concentrations of Zn and Cu in the extract were low and reached 1.39 ± 0.04 and 0.803 ± 0.065 μg/L, respectively [13].

According to the producer’s declaration, the extract also contained sugar, sugar alcohols (sorbitol, parasorboside), triterpenes, phytosterols, carotenoids, minerals, and vitamins. Moreover, vitamins from group B (B_1_, B_2_, B_3_, B_5_, and B_6_), vitamins C, E, and K, β-carotene, β-cryptoxanthin, violaxanthin, dietary fiber, tannins, organic acids (l-malic acid, citric acid), carbohydrates, proteins, as well as calcium, magnesium, and iron [1,9] are present in chokeberries. The concentration of Cd in the 0.1% AE was <0.05 μg/L [10].

### 4.3. Experimental Design

The experiment was divided into two stages. In the first stage, the ability of the 0.05% and 0.1% AE to form complexes with Zn^2+^ at the concentrations of 0.01, 0.1, and 3 mM and Cu^2+^ at the concentrations of 0.01, 0.05, and 0.5 mM was evaluated at pH 2, 5.5, and 8. In the second stage, the abilities of the main *Aronia* polyphenols at the concentrations corresponding to their concentrations in the 0.1% AE to complex these ions were evaluated. C3G (the most abundant anthocyanin in the extract) and CA were used at the concentrations of 0.008% and 0.007%, respectively, determined by us in the 0.1% AE (Table 2) [11]. Other *Aronia* polyphenols were used at the following concentrations: NCA—0.007%, (−)-epicatechin—0.013%, (+)-catechin— 0.013%, Q—0.002%, and K—0.002%.

In order to obtain the solutions containing Zn^2+^ at the concentrations of 0.01, 0.1, and 3 mM and Cu^2+^ at the concentrations of 0.01, 0.05, and 0.5 mM, the solutions containing 1.36, 13.6, and 408 mg ZnCl_2_/10 mL, respectively, as well as 1.6, 8, and 80 mg CuSO_4_/10 mL, respectively, were prepared in methanol:H_2_O (7:3) directly before the experiment. The concentrations of anhydrous salts of Zn and Cu (ZnCl_2_ and CuSO_4_) were calculated to achieve needed concentrations of Zn^2+^ and Cu^2+^, taking into account these salts’ solubility in H_2_O at room temperature (432 g ZnCl_2_/100 mL and 23 g CuSO_4_/100 mL; CuSO_4_ is suitably soluble in methanol, and this compound dissolved well in the mixture of methanol:H_2_O at the ratio of 7:3) and their molar mass.

The solutions of the AE, particular polyphenolic compounds, and EDTA in methanol:H_2_O (7:3) were prepared directly before the experiment. The needed pH values (checked with the use of buffer solutions and pH meter; Mettler Toledo, Schwarzenbach, Switzerland) were obtained by adding appropriate amounts of 5% HCl or 0.1% NaOH to stock solutions of the AE, particular polyphenols, EDTA, and bioelements. Assessment of the stability of the 0.1% AE revealed that the extract, at all used pH, is stable during the first 24 h after preparation (Appendix A—Assessment of the stability of the extract from the berries of *Aronia melanocarpa* L. (AE)).

### 4.4. The Rule of the Method of Estimation of Metal Ion Complexation by AE and Polyphenolic Compounds

The complexation abilities of the 0.1% and 0.05% AE, as well as of particular polyphenols towards Zn^2+^ and Cu^2+^ were evaluated based on shifts between the maximum absorption of solutions before an addition of ZnCl_2_ or CuSO_4_ and the maximum absorption of the respective solutions after addition of these salts [16,34,35]. The shift of the maximum absorption observed after the addition of the solutions containing Zn^2+^ or Cu^2+^ to the solutions with the AE (0.05% and 0.1%) or polyphenolic compounds indicates that the process of chelation of the ions occurred. A lack of the shift of the maximum absorption means that the process of chelation of metal ions by the used solution did not occur. As a positive control, EDTA solutions (with and without Zn^2+^ and Cu^2+^) were used at the concentrations analogous to the investigated concentrations of the AE (0.05% and 0.1%) and particular polyphenolic compounds (0.002%, 0.007%, 0.008%, and 0.013%).

Three hundred microliters of the solutions of AE, particular polyphenolic compounds, or EDTA, at appropriate concentrations, were mixed with 300 µL of the prepared solutions of ZnCl_2_ or CuSO_4_ in a quartz cuvette, and then the UV–Vis absorption spectrum of the mixture was recorded. Absorption spectra of the solutions were recorded within the range of 200–800 nm using a Specord 50 Plus spectrophotometer (Analytik, Jena, Germany). Each reaction was repeated three times to check the coherence of samples. Owing to the fact that the results obtained from each of the three independent models containing the AE, particular polyphenolic compounds, or EDTA with or without Zn^2+^ and Cu^2+^ were repeatable, the results have been presented for only one representative sample from each model.

## Figures and Tables

**Figure 1 molecules-25-01507-f001:**
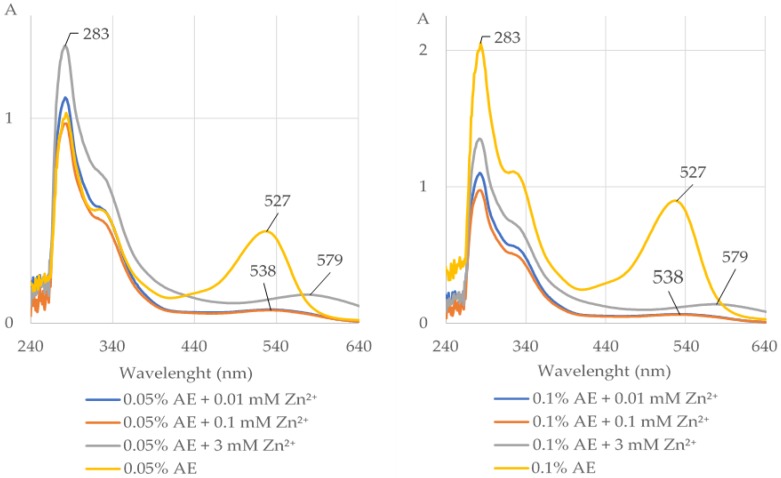
The UV–Vis absorption spectra (with indicated maximum absorption) of the extract from *Aronia melanocarpa* L. berries (0.05% and 0.1% AE) alone and after addition of divalent ions of zinc (Zn^2+^) at the concentrations of 0.01, 0.1, and 3 mM at pH 5.5. The shift of the maximum absorption after addition of Zn^2+^ to the AE indicates that the extract chelated these ions. A, absorbance.

**Figure 2 molecules-25-01507-f002:**
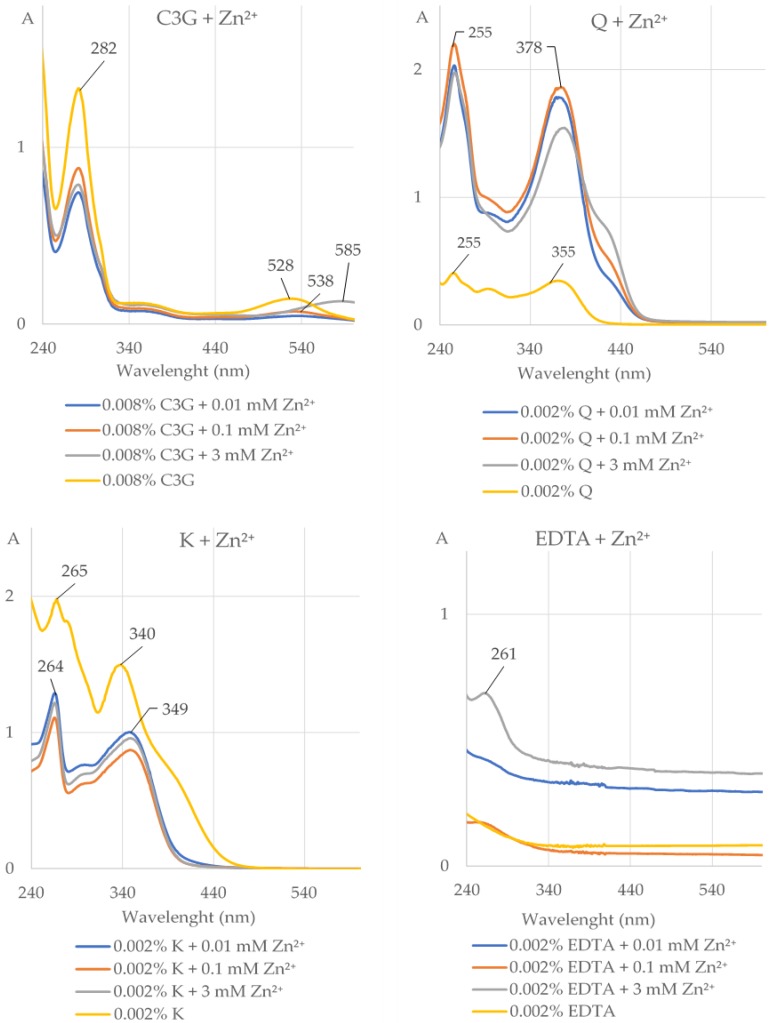
The UV–Vis absorption spectra (with indicated maximum absorption) of cyanidin 3-*O-β*-galactoside (0.008% C3G), quercetin (0.002% Q), kaempferol (0.002% K), and ethylenediaminetetraacetic acid (0.002% EDTA) alone and after addition of divalent ions of zinc (Zn^2+^) at the concentrations of 0.01, 0.1, and 3 mM at pH 5.5. The shift of the maximum absorption after addition of Zn^2+^ to the solution of the investigated compound indicates that this compound chelated these ions. A, absorbance.

**Figure 3 molecules-25-01507-f003:**
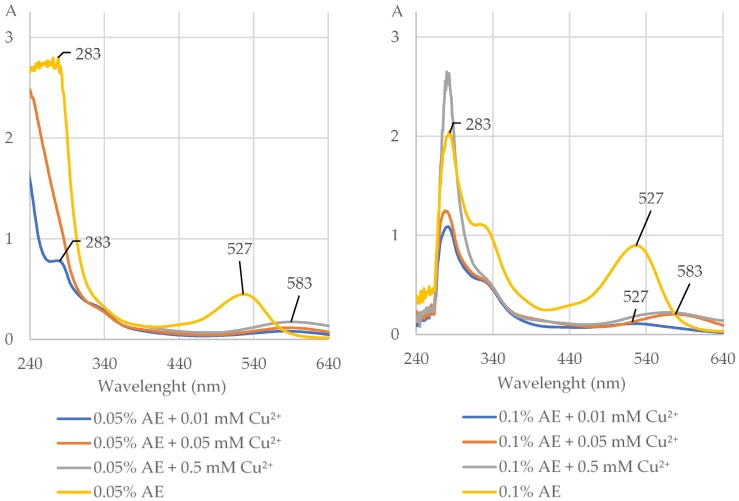
The UV–Vis absorption spectra (with indicated maximum absorption) of the extract from *Aronia melanocarpa* L. berries (0.05% and 0.1% AE) alone and after addition of divalent ions of copper (Cu^2+^) at the concentrations of 0.01, 0.05, and 0.5 mM at pH 5.5. The shift of the maximum absorption after addition of Cu^2+^ to the AE indicates that the extract chelated these ions. A, absorbance.

**Figure 4 molecules-25-01507-f004:**
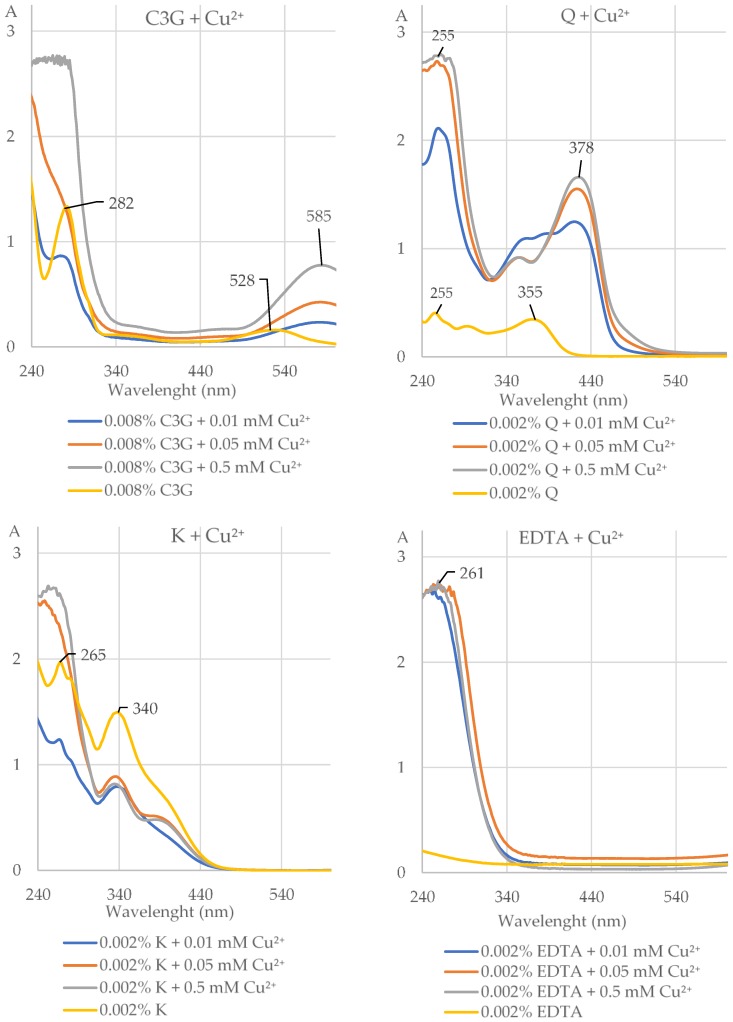
The UV–Vis absorption spectra (with indicated maximum absorption) of cyanidin 3-*O*-*β*-galactoside (0.008% C3G), quercetin (0.002% Q), kaempferol (0.002% K), and ethylenediaminetetraacetic acid (0.002% EDTA) alone and after addition of divalent ions of copper (Cu^2+^) at the concentrations of 0.01, 0.05, and 5 mM at pH 5.5. The shift of the maximum absorption after addition of Cu^2+^ to the solution of the investigated compound indicates that this compound chelated these ions. A, absorbance.

**Figure 5 molecules-25-01507-f005:**
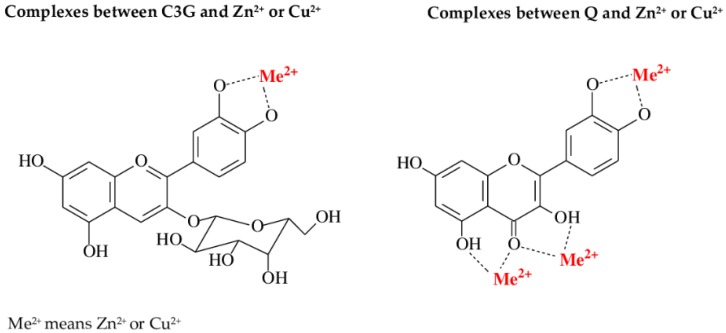
The possible complexes between cyanidin 3-*O-β*-galactoside (C3G) and quercetin (Q) and divalent ions of zinc (Zn^2+^) and copper (Cu^2+^).

**Table 1 molecules-25-01507-t001:** Summarizing the results of the evaluation of complexation of divalent ions of zinc (Zn^2+^) at the concentrations of 0.01, 0.1, and 3 mM and copper (Cu^2+^) at the concentrations of 0.01, 0.05, and 0.5 mM by 0.1% extract from *Aronia melanocarpa* L. berries (AE) and the main polyphenolic compounds present in the extract, as well as by ethylenediaminetetraacetic acid (EDTA). ^1, 2.^

AE, Polyphenolic Compound or EDTA	pH 2		pH 5.5		pH 8	
	Zn^2+^	Cu^2+^	Zn^2+^	Cu^2+^	Zn^2+^	Cu^2+^
0.1% AE	−	−	+	+	−	−
0.008% C3G	−	−	+	+	−	−
0.002% Q	−	−	+	+	−	−
0.002% K	−	−	+	−	−	−
0.007% CA	−	−	−	−	−	−
0.007% NCA	−	−	−	−	−	−
0.013% (+)-catechin	−	−	−	−	−	−
0.013% (−)-epicatechin	−	−	−	−	−	−
0.1% EDTA	−	−	+	+	−	−
0.013% EDTA	−	−	+	+	−	−
0.008% EDTA	−	−	+	+	−	−
0.007% EDTA	−	−	+	+	−	−
0.002% EDTA	−	−	+	+	−	−

^1^ The same effect was noted at all concentrations of Zn^2+^ (0.01, 0.1, and 3 mM). ^2^ The same effect was noted at all concentrations of Cu^2+^ (0.01, 0.05, and 0.5 mM). C3G, cyanidin 3-*O*-*β*-galactoside; Q, quercetin; K, kaempferol; CA, chlorogenic acid; NCA, neochlorogenic acid. +, complexation; −, lack of complexation.

**Table 2 molecules-25-01507-t002:** The concentrations of polyphenolic compounds in the 0.1% extract from *Aronia melanocarpa* L. berries (adapted from [11]).

Polyphenolic Compounds	Concentration [μg/L]
Total polyphenols	612.40 ± 3.33 ^3^
Total anthocyanins	202.28 ± 1.28
Total proanthocyanidins	129.87 ± 1.12
Total phenolic acids	110.92 ± 0.89
Total flavonoids	21.94 ± 0.98
Cyanidin 3-*O*-*β*-galactoside	80.07 ± 1.05
Cyanidin 3-*O*-*α*-arabinoside	33.21 ± 0.01
Cyanidin 3-*O*-*β*-glucoside	3.68 ± 0.01
Chlorogenic acid	68.32 ± 0.08

^3^ Data are mean ± standard error (*n* = 3).

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
