# Peer review of "Estimation of the Chelating Ability of an Extract from Aronia melanocarpa L. Berries and Its Main Polyphenolic Ingredients Towards Ions of Zinc and Copper"

_molecules, 2020, doi:10.3390/molecules25071507_

Round 1

Reviewer 1 Report

Brzóska and co workers described the chelating ability of an extract of Aronia  ragerding divalen ions of copper and zinc.

I believe that the topic is very interesting, but the experiments performed are not accurate enough to confirm these properties exhaustively. All data collected are based on a single technique (U.V.). I would recommend adding some additional evidence collected with different techniques.

Moreover, the graphs and tables in the manuscript are not easy to read.

In Figure 1 and Figure 2: which pH do they use? In what solvent (probably water)? Has a buffer been used? What kind?

On page 3 line 104-105 The authors speak of the absorption of C3G at 528 nm but no figure shows it.

Figure 2: what does it represent? Is it the simple UV spectrum or is it the difference between the spectrum of the free compound and the complexed one? When EDTA is shown there, seems to be no absorption but in Tab. 3 we talk about 219, 261, 356, 745 nm.

They speak about a different complexation ability when the pH change. Have stability tests of compounds (without metal) ever been carried out at pH 2, 5.5 and 8?

The pH influences the protonation / deprotonation of the -OH groups changing their ability to bind metal ions and sometimes decrease the stability of the polyphenolic compounds (especially at pH 8).

For these reasons I recommend accepting this work only after major revisions

Author Response

Response to the remarks of Reviewer 1

 Manuscript molecules-734936

The authors would like to thank the Reviewer 1 for valuable comments towards improving the manuscript molecules-734936. We deeply appreciate the Reviewer time and assistance. All comments and suggestions of the Reviewer have been carefully read and taken into consideration during preparation of the revised version. All changes made in the manuscript in response to the Reviewers’ remarks have been written in red and highlighted.

Below we answer the comments raised by the Reviewer 1.

Response to the Review Report Form

Does the introduction provide sufficient background and include all relevant references?

Reviewer: Yes

Authors’ response: The authors thank the Reviewer for the positive opinion on the Introduction section.

Is the research design appropriate?

Reviewer: Must be improved

Authors’ response: Some corrections have been made to improve the description of the research design.

Are the methods adequately described?

Reviewer: Must be improved

Authors’ response: Numerous corrections have been made in the Materials and Methods section to improve adequateness of the methods description.

Are the results clearly presented?

Reviewer: Must be improved

Authors’ response: Numerous corrections have been made to improve the form of results presentation.

Are the conclusions supported by the results?

Reviewer: Must be improved

Authors’ response: Conclusions have been corrected to make them better reflecting the results.

Point-by-point response to the Reviewer 1 Comments and Suggestions for Authors

Reviewer comment

Brzóska and co workers described the chelating ability of an extract of Aronia regarding divalent ions of copper and zinc.

I believe that the topic is very interesting, but the experiments performed are not accurate enough to confirm these properties exhaustively. All data collected are based on a single technique (U.V.). I would recommend adding some additional evidence collected with different techniques.

Authors’ response

The authors are aware that demonstrating the ability of the extract from chokeberries and its chosen polyphenolic ingredients to chelate divalent ions of zinc and copper with the use of other techniques would be an important confirmation of our findings; however, at this moment we have no possibility of carrying out such studies. Performing such study with the use of different techniques would require planning a completely new experiment and needs financial support, time and availability of the same samples of aronia extract and polyphenolic compounds. That is why, at this moment we are unable to perform such study and have to limit to a single method. The used by us spectrophotometric method, based on the shift of the maximum of absorption of the solution of aronia extract or polyphenolic compound before an addition of ions of zinc or copper, and the maximum of absorption of the respective solutions after addition of these ions, is also used by other authors. Planning our study we have checked that evaluation of metal-chelating properties of polyphenolic compounds is commonly done by means of UV-VIS absorption spectroscopy and that the results of such investigations were and are published in leading scientific journals (for example Andjelković et al., Food Chem. 2006, 98, 23-31; Zhang et al., Food Chem. 2019, 293, 66-73; Fedenko et al., J. Plant Physiol., http://dx.doi.org/doi:10.1016/j.jplph.2017.02.001). Moreover, it has been revealed that results of the investigation of metals complexation with spectrophotometric methods are in agreement with the results of investigations involving mass spectrometry (Fernandez et al., J. Inorg. Biochem., 1998, 71, 93-98).

The present study is our first research, and the first in general, aimed to estimate the complexing ability of an extract from aronia berries towards divalent ions of zinc and copper. We wanted first of all to reveal whether the extract is capable of binding ions of zinc and copper to explain the findings of our in vivo study on the impact of aronia extract on the body status of zinc and copper under and without exposure to cadmium. That is why, we have used the spectrophotometric method because it provides response on the ability to metal ions complexation or its lack. The fact of using aronia extract at two concentrations (0.1% and 0.05%) and revealing that the extract is capable of binding ions of zinc and copper at both concentrations at pH 5.5 confirm its chelating ability. Moreover, the fact that cyanidin 3-O-β-galactoside and quercetin, being the extract polyphenolic ingredients, bound Zn2+ and Cu2+ (pH 5.5) provides further confirmation of the extract chelating ability. That is why, we think that our conclusion regarding the complexing abilities of the extract towards divalent ions of zinc and copper is entitled. Of course, revealing in our study that aronia extract may complex divalent ions of zinc and copper indicates the need of planning further study to more detailed investigation of the efficiency of binding particular elements and the stability of these complexes. Our results presented in the manuscript submitted for consideration for publication in the Special Issue of Molecules “Chelating Agents Towards Bioavailable Metal Ions” may be useful for explanation of the results of our earlier studies, concerning the influence of the extract from Aronia melanocarpa L. berries on the body status of zinc and copper. First of all our study is the first report indicating the complexing properties of aronia extract towards zinc and copper.

The authors entirely agree with the Reviewer opinion that the fact of the lack of confirmation of our results by other methods reduces the enthusiasm resulting from our findings. This limitation of our study has been mentioned in the Discussion section. It has been explained in the Discussion section that the main limitation of our study is the fact that the chelating properties of the aronia extract and particular polyphenolic compounds towards Zn2+ and Cu2+ or their lack were investigated only with the use of one method and they have not been estimated and confirmed by other methods. However, because the spectrophotometric method is used to estimate the complexing ability of various compounds towards ions of metals, our findings cannot be considered insignificant. The present study may constitute a basis for further studies in this matter with the use of different techniques. The complexing properties of aronia extract have not been investigated until now, but our finding of complexation of Zn2+ and Cu2+ by the extract and chosen its polyphenolic ingredients show the need of further study on this issue. In the future, we would like to continue our study with the use of other techniques aimed to more complete investigation the chelating abilities of the aronia extract and its ingredients. Unfortunately, at this moment we have no possibility to perform such studies and include their findings in the current paper.

Reviewer comment

Moreover, the graphs and tables in the manuscript are not easy to read.

In Figure 1 and Figure 2: which pH do they use? In what solvent (probably water)? Has a buffer been used? What kind?

Authors’ response

Figures and tables have been corrected to make them more readable.

Figures 1 and 2 present the chelating ability of the extract from Aronia melanocarpa L. berries (0.05 and 0.1% AE) and its chosen polyphenolic ingredients towards divalent ions of zinc at pH 5.5, whereas Figures 3 and 4 present the same regarding to copper. In the figure legends, the pH was not provided; however, it was underlined in the manuscript text (the Results section) that aronia extract and its polyphenolic ingredients chelated ions of copper and zinc only at pH 5.5. In the revised manuscript, the information that Figures 1 - 4 present the chelating ability of the extract from Aronia melanocarpa L. berries (0.05 and 0.1% AE) and its chosen polyphenolic ingredients towards divalent ions of zinc or copper at pH 5.5 has been provided. All UV–VIS absorption spectra were re-analyzed and completely new Figures 1 - 4, presenting the UV–VIS absorption spectra (with indicated maximum of absorption) of the solutions of aronia extract, chosen polyphenolic compounds, and ethylenediaminetetracetic acid alone, as well as after addition of Zn2+ or Cu2+ have been prepared. In aim to make the figures more readable, maxima absorption have been marked and it has been explained that the shift of the maximum of absorption after addition of metal ions to the solution of aronia extract or investigated polyphenolic compound indicates that this solution/compound chelated these ions.

Taking into account comment of the Reviewer 4, Tables 1 and 2, presenting the effect of divalent ions of zinc and copper at the studied concentrations on wavelength of absorption of the extract from Aronia melanocarpa L. berries (at the concentration of 0.05% and 0.1%), 0.1% ethylenediaminetetracetic acid, and the main polyphenolic compounds at various pH have been removed from the main text and moved into Supplementary data (Table S2 and Table S3). Moreover in Table S3, presenting the effect of Zn2+ and Cu2+ at the studied concentrations on wavelength of absorption of the main polyphenolic compounds present in the 0.1% extract from Aronia melanocarpa L. berries at pH 5.5, the numerical values of the shift of the maximum of absorption (positive value of the shift reflects metal complexation) observed after the addition of the solutions containing Zn2+ or Cu2+ to the solutions of polyphenolic compounds have been provided.

The solutions of the aronia extract, particular polyphenolic compounds, and ethylenediaminetetracetic acid, as well as zinc and copper were prepared (directly before the use) in methanol : water (7 : 3). The data about the used solvents are provided in the Materials and Methods section. No buffer solutions were added. Buffer solutions of pH 4.01 and 10.01 were used to check pH of particular solutions. The performed by us spectrophotometric assessment of the stability of 0.1% extract from the berries of Aronia melanocarpa L. at pH 2, 5.5, and 8 after preparation (1h), as well as after 24h, 48h, and 72h revealed that these solutions were stable at all pH during the first 24h after preparation. Data on the stability assessment have been presented as Supplementary Material – Assessment of the stability of the extract from the berries of Aronia melanocarpa L. (AE). Because all solutions were prepared immediately before use, they were stable during the study.

Reviewer comment

On page 3 line 104-105 The authors speak of the absorption of C3G at 528 nm but no figure shows it.

Authors’ response

The authors would like to thank the Reviewer very much for this remark. The figure presenting a lack of complexation of Zn2+ by chlorogenic acid was mistakenly used instead of the figure presenting Zn2+ complexation by cyanidin 3-O-β-galactoside. In the revised manuscript, appropriate correction has been made. As it was above mentioned, all UV-VIS spectra were re-analyzed and new Figures 1 - 4 have been prepared and appropriate corrections have been made in the text.

We would like to apologize for this mistake.

Reviewer comment

Figure 2: what does it represent? Is it the simple UV spectrum or is it the difference between the spectrum of the free compound and the complexed one? When EDTA is shown there, seems to be no absorption but in Tab. 3 we talk about 219, 261, 356, 745 nm.

Authors’ response

Figure 2 represents the UV–VIS absorption spectra (with indicated maximum of absorption) of the solutions of chosen polyphenolic compounds and ethylenediaminetetracetic acid alone, as well as after addition of Zn2+ at pH 5.5. Under each of the four sets of figures a legend explaining that the spectra of cyanidin 3-O-β-galactoside, quercetin, kaempferol, and ethylenediaminetetraacetic acid without and with Zn2+ are presented. As it was above mentioned, all UV-VIS spectra, including these of ethylenediaminetetracetic acid, alone, as well as after addition of Zn2+ at pH 5.5, were re-analyzed and appropriate corrections have been made in the manuscript. New Figures presenting UV-VIS spectra of the ethylenediaminetetracetic acid alone and after addition of Zn2+ or Cu2+ (Figure 2 and Figure 4) have been prepared. Data presented in figures are consistent with these presented as Supplementary Tables S2 and S6).

Reviewer comment

They speak about a different complexation ability when the pH change. Have stability tests of compounds (without metal) ever been carried out at pH 2, 5.5 and 8?

The pH influences the protonation / deprotonation of the -OH groups changing their ability to bind metal ions and sometimes decrease the stability of the polyphenolic compounds (especially at pH 8).

Authors’ response

We are aware that pH may influence the stability of the polyphenolic compounds and thus we have checked the stability of the extract from the berries of Aronia melanocarpa at all used values of pH (2, 5.5 and 8); however, the findings have not been presented in the previous version of the manuscript. Taking into account the above comment of the Reviewer 1, in the Materials and Methods section it has been stated that extract from the berries of Aronia melanocarpa at all used values of pH was stable within 24 h. As it was above mentioned, the findings have been presented as the Supplementary data (Supplementary Material – Assessment of the stability of the extract from the berries of Aronia melanocarpa L. (AE)). In the Materials and Methods section it is stated that all solutions were prepared immediately before use. Thus, all solutions were stable during the study.

Reviewer comment

For these reasons I recommend accepting this work only after major revisions.

Authors’ response

The Authors have made efforts to revise the manuscript according the recommendations of the Reviewer 1. If any additional corrections or explanations are needed we are ready to do them immediately.

Apart from the above-described changes made in the manuscript in response to the remarks of the Reviewer 1, some corrections have been introduced in response to the remarks of the Reviewer 4. Because two other Reviewers have no any remarks, all changes made in the manuscript resulted from the remarks of the Reviewer 1 or Reviewer 4.

Once again the authors would like to thank the Reviewer 1 for contribution in improving the manuscript.

Yours faithfully,

Malgorzata Brzóska

on behalf of all co-authors

Reviewer 2 Report

Present research is very interesting and innovative. The manuscript is very well structured. The information provided is very useful for the nutrition science and human diet. The article will be in high interest to the readers. The introduction provides sufficient background and describes clear the main goal of the present research. The references used are sufficient in number and cover the available information on the topic. The available data are analyzed, interpreted and critical reviewed in detail and correctly. Conclusion highlights the main aspects on the study.

Author Response

Response to the remarks of Reviewer 2

Manuscript molecules-734936

Response to the Review Report Form

Does the introduction provide sufficient background and include all relevant references?

Reviewer: Yes

Authors’ response: The authors thank the Reviewer for the positive opinion on the Introduction section.

Is the research design appropriate?

Reviewer: Yes

Authors’ response: The authors thank the Reviewer for this opinion. Taking into account comments of the Reviewer 1 and Reviewer 4, description of the research design has been improved.

Are the methods adequately described?

Reviewer: Yes

Authors’ response: The authors thank the Reviewer for this opinion. Taking into account comments of the Reviewer 1 and Reviewer 4, numerous corrections have been made in the Materials and Methods section to improve adequateness of the methods description.

Are the results clearly presented?

Reviewer: Yes

Authors’ response: The authors thank the Reviewer for this opinion; however, in response to remarks of the Reviewer 1 and Reviewer 4 some corrections have been made to improve the form of results presentation.

Are the conclusions supported by the results?

Reviewer: Yes

Authors’ response: The authors thank the Reviewer for this opinion. Taking into account comments of the Reviewer 1 and Reviewer 4, conclusions have been corrected to make them better reflecting the results.

Comments and Suggestions for Authors

Reviewer comment

Present research is very interesting and innovative. The manuscript is very well structured. The information provided is very useful for the nutrition science and human diet. The article will be in high interest to the readers. The introduction provides sufficient background and describes clear the main goal of the present research. The references used are sufficient in number and cover the available information on the topic. The available data are analyzed, interpreted and critical reviewed in detail and correctly. Conclusion highlights the main aspects on the study.

Authors’ response

The authors thank the Reviewer for this opinion; however, some changes have been made in the manuscript in response to the remarks of the Reviewer 1 and Reviewer 4.

Once again the authors would like to thank the Reviewer 2 for preparing the review of our manuscript.

Yours faithfully,

Malgorzata Brzóska

on behalf of all co-authors

Reviewer 3 Report

Ms. Ref. No.: Manuscript ID: molecules-734936 presents the “Estimation of the chelating ability of an extract from Aronia melanocarpa L. berries and its main polyphenolic ingredients towards ions of zinc and copper”.

This is an interesting research manuscript. The abstract covers the information presented in the manuscript and the key words are suitable for its presentation. The MS has an appropriate structure, the materials and methods are quite informative to allow replication of the experiment, the results are clearly presented, the tables and figures are all necessary, complete and clearly presented. The references are adequate.

Author Response

Response to the remarks of Reviewer 3

Manuscript molecules-734936

Response to the Review Report Form

Does the introduction provide sufficient background and include all relevant references?

Reviewer: Yes

Authors’ response: The authors thank the Reviewer for the positive opinion on the Introduction section.

Is the research design appropriate?

Reviewer: Yes

Authors’ response: The authors thank the Reviewer for this opinion. Taking into account comments of the Reviewer 1 and Reviewer 4, description of the research design has been improved.

Are the methods adequately described?

Reviewer: Yes

Authors’ response: The authors thank the Reviewer for this opinion. Taking into account comments of the Reviewer 1 and Reviewer 4, numerous corrections have been made in the Materials and Methods section to improve adequateness of the methods description.

Are the results clearly presented?

Reviewer: Yes

Authors’ response: The authors thank the Reviewer for this opinion; however, in response to remarks of the Reviewer 1 and Reviewer 4 some corrections have been made to improve the form of results presentation.

Are the conclusions supported by the results?

Reviewer: Yes

Authors’ response: The authors thank the Reviewer for this opinion. Taking into account comments of the Reviewer 1 and Reviewer 4, conclusions have been corrected to make them better reflecting the results.

Comments and Suggestions for Authors

Reviewer comment

Ms. Ref. No.: Manuscript ID: molecules-734936 presents the “Estimation of the chelating ability of an extract from Aronia melanocarpa L. berries and its main polyphenolic ingredients towards ions of zinc and copper”.

This is an interesting research manuscript. The abstract covers the information presented in the manuscript and the key words are suitable for its presentation. The MS has an appropriate structure, the materials and methods are quite informative to allow replication of the experiment, the results are clearly presented, the tables and figures are all necessary, complete and clearly presented. The references are adequate.

Authors’ response

The authors thank the Reviewer for this opinion; however, some changes have been made in the manuscript in response to the remarks of the Reviewer 1 and Reviewer 4.

Once again the authors would like to thank the Reviewer 3 for preparing the review of our manuscript.

Yours faithfully,

Malgorzata Brzóska

on behalf of all co-authors

Reviewer 4 Report

The manuscript reports on chelating ability of Aronia extracts and several phenolic compounds towards Cu(II) and Zn(II) ions. Unfortunately, extent of the research is insufficient, the applied methodology is not suitable and the results are of a very limited value. Therefore, I suggest rejecting the manuscript.

The presented research is not a relevant research. Measurement of few UV-VIS spectra is insufficient for characterization of the studied systems and for making any conclusions. Complexation of metal ions with studied model compounds should be studied more in detail. Ligand dissociation constants should be determined from UV-VIS spectra or from potentiometry. The absorption spectra of complexes should be measured as function of ligand or metal ion concentration in a broad range (typically 10 to 20 pints at 0:1 to 20:1 ligand-to-metal ratio) and in buffered solutions to maintain pH. It will give an indication on stoichiometry and stability of the studied complexes. The data should be presented in the form of whole spectra and in the form of absorbance at maximum as a function of metal-to-ligand molar ratio. Conditional and thermodynamic stability constants of the studied complexes should be calculated.

The presented tables containing maxima of absorption wavelengths are useless.

Maximum of the absorption band is not sufficient support to make the presented conclusions on complexation of Cu(II) and Zn(II) with Aronia extract.

The current form of chapter Discussion is maybe suitable for a food science journal. There is only very limited discussion on the chemical behavior of the studied ligands and complexes. Many of the conclusions are speculations that are not supported by experimental evidence.

The concept of chapter Materials and Methods is not suitable. The used procedures should be described in details but without discussion in this chapter. All discussion should be moved to corresponding chapter Discussion.

The presented form is of chapter Materials and Methods difficult to read and follow. The concentrations should be given as mmol/L, amounts of ligands and metals should be given in milligrams and/or milimols. Volumes of all solutions should be given.

Author Response

Response to the remarks of Reviewer 4

Manuscript molecules-734936

The authors would like to thank the Reviewer 4 for valuable comments towards improving the manuscript molecules-734936. We deeply appreciate the Reviewer time and assistance. All comments and suggestions of the Reviewer have been carefully read and taken into consideration during preparation of the revised version. All changes made in the manuscript in response to the Reviewers’ remarks have been written in red and highlighted.

Below we answer the comments raised by the Reviewer 4.

Response to the Review Report Form

Does the introduction provide sufficient background and include all relevant references?

Reviewer: Yes

Authors’ response: The authors thank the Reviewer for the positive opinion on the Introduction section.

Is the research design appropriate?

Reviewer: Must be improved

Authors’ response: Some corrections have been made to improve the description of the research design.

Are the methods adequately described?

Reviewer: Must be improved

Authors’ response: Numerous corrections have been made in the Materials and Methods section to improve adequateness of the methods description.

Are the conclusions supported by the results?

Reviewer: Must be improved

Authors’ response: Conclusions have been corrected to make them better reflecting the results. Conclusions are drawn more carefully.

Point-by-point response to the Reviewer 4 Comments and Suggestions for Authors

Reviewer comment

The manuscript reports on chelating ability of Aronia extracts and several phenolic compounds towards Cu(II) and Zn(II) ions. Unfortunately, extent of the research is insufficient, the applied methodology is not suitable and the results are of a very limited value. Therefore, I suggest rejecting the manuscript.

The presented research is not a relevant research. Measurement of few UV-VIS spectra is insufficient for characterization of the studied systems and for making any conclusions. Complexation of metal ions with studied model compounds should be studied more in detail. Ligand dissociation constants should be determined from UV-VIS spectra or from potentiometry. The absorption spectra of complexes should be measured as function of ligand or metal ion concentration in a broad range (typically 10 to 20 pints at 0:1 to 20:1 ligand-to-metal ratio) and in buffered solutions to maintain pH. It will give an indication on stoichiometry and stability of the studied complexes. The data should be presented in the form of whole spectra and in the form of absorbance at maximum as a function of metal-to-ligand molar ratio. Conditional and thermodynamic stability constants of the studied complexes should be calculated.

Authors’ response

The authors would like to thank the Reviewer 4 for all valuable comments regarding the methodology of studying the complexing ability towards Zn2+ and Cu2+. We are aware that demonstrating the ability of the extract from chokeberries and the investigated polyphenolic ingredients of the extract to chelate divalent ions of zinc and copper with other methods (e.g. potentiometry, mass spectrometry) would be an important confirmation of their complexing abilities; however, the used by us spectrophotometric method is a method recognized as appropriate to study the complexing ability towards ions of metals. This method was and is used by other authors and the results of such investigations are published in leading scientific journals (for example Andjelković et al., Food Chem. 2006, 98, 23-31; Zhang et al., Food Chem. 2019, 293, 66-73; Fedenko et al., J. Plant Physiol., http://dx.doi.org/doi:10.1016/j.jplph.2017.02.001). Planning our study we have taken into account also the fact that it has been revealed that results of the investigation of metals complexation with spectrophotometric method are in agreement with the results of investigations involving mass spectrometry (Fernandez et al., J. Inorg. Biochem., 1998, 71, 93-98).

We absolutely agree that determination of other parameters such as for example ligand dissociation constants would provide very useful data on the chemistry of metals complexation by the aronia extract, but our study was aimed to recognize whether this extract may complex divalent ions of zinc and copper in general. Our study is the first research aimed to estimate whether the extract is capable of binding ions of zinc and copper to explain the findings of our recent in vivo study on the impact of aronia extract on the body status of these bioelements under and without exposure to cadmium. We wanted first of all to reveal whether the extract is capable of binding Zn2+ and Cu2+. Revealing in our study that aronia extract may complex divalent ions of zinc and copper indicates the need of planning further study to investigate the efficiency of binding and the stability of these complexes. The present study may constitute a basis for further studies in this matter with the use of different techniques. In the future, we would like to continue our study with the use of other techniques aimed to more complete investigation the chelating abilities of the aronia extract and its ingredients. Planning the study we will take into account all very valuable recommendations of the Reviewer 4 as well as available literature data (including the very advanced study reported by Zhang et al., Food Chem. 2019, 293, 66-73).

At the stage of preparation of the revised version of the manuscript we were unable to perform additional studies. Performing the study with the use of different techniques would require to plan and conduct a completely new study. Such study needs financial support, time and availability of the same samples of aronia extract and polyphenolic compounds. That is why, at this moment we are unable to perform such study and have to limit to a single method. This limitation of our study related to its methodology has been mentioned in the Discussion section. It has been explained that the main limitation of our study is the fact that the chelating properties of the aronia extract and particular polyphenolic compounds towards Zn2+ and Cu2+ or their lack were investigated only with the use of one method and they have not been estimated and confirmed by other methods, as well as that the chemical nature (e.g. stability, ligand dissociation constants) of the formed complexes has not been assessed.

The Reviewer suggested that the absorption spectra of complexes should be measured as a function of metal ion concentrations. It our study 3 various concentrations of Zn2+ and Cu2+ have been used and there were no differences in the ability of the aronia extract and particular polyphenolic compounds to complex these elements dependent on their concentration.

Because aronia extract is composed of numerous ingredients it seems impossible to present the results in the form of absorbance at maximum as a function of metal-to-ligand molar ratio.

Reviewer comment

The presented tables containing maxima of absorption wavelengths are useless.

Authors’ response

Taking into account this comment Tables 1 and 2, presenting the effect of divalent ions of zinc and copper at the studied concentrations on wavelength of absorption of the extract from Aronia melanocarpa L. berries (at the concentration of 0.05% and 0.1%), 0.1% ethylenediaminetetracetic acid, and the main polyphenolic compounds at various pH have been removed from the main text and moved into Supplementary data. Moreover in Table S3, presenting the effect of Zn2+ and Cu2+ at the studied concentrations on wavelength of absorption of the main polyphenolic compounds present in the 0.1% extract from Aronia melanocarpa L. berries at pH 5.5, the numerical values of the shift of the maximum of absorption (positive value of the shift reflects metal complexation) observed after the addition of the solutions containing Zn2+ or Cu2+ to the solutions of polyphenolic compounds have been provided.

Thank you very much for this suggestion. These data are not necessary in the main text; however, they may be interesting for researchers who will plan their own studies on the complexing abilities of polyphenols regards to bioelements and thus we have decided to present them as the Supplementary data.

Reviewer comment

Maximum of the absorption band is not sufficient support to make the presented conclusions on complexation of Cu(II) and Zn(II) with Aronia extract.

Authors’ response

We are aware that various methods may be used for examination of the complexing abilities. However, taking into account available literature data and the fact that the aim of our study was to recognize first of all whether the extract is capable of binding ions of zinc and copper to explain the findings of our recent in vivo study on the impact of aronia extract on the body status of zinc and copper under and without exposure to cadmium. The used by us spectrophotometric method of evaluation of metal-chelating properties, which is based on the shift of the maximum of absorption of the investigated solution (aronia extract or polyphenolic compound) before an addition of ions of metals (Zn2+ and Cu2+) and the maximum of absorption of the respective solution after addition of these ions, is nowadays used also by other authors (Zhang et al., Food Chem. 2019, 293, 66-73; Fedenko et al., J. Plant Physiol., http://dx.doi.org/doi:10.1016/j.jplph.2017.02.001).

Our study is the first research aimed to estimate the complexing ability of an extract from aronia berries towards divalent ions of zinc and copper. We wonted first of all to reveal whether the extract is capable of binding ions of zinc and copper to explain the findings of our recent in vivo study on the impact of aronia extract on the body status of zinc and copper under and without exposure to cadmium. The fact of using aronia extract at two concentrations (0.1% and 0.05%) and revealing that the extract is capable of binding ions of zinc and copper at both concentrations at pH 5.5 confirm its chelating ability. Moreover, the fact that cyanidin 3-O-β-galactoside and quercetin, being the extract polyphenolic ingredients, bound Zn2+ and Cu2+ (pH 5.5) provides further confirmation of the extract chelating ability. That is why; we think that our conclusion regarding the complexing abilities of the extract towards divalent ions of zinc and copper is entitled. Our results presented in the manuscript submitted for consideration for publication in the Special Issue of Molecules “Chelating Agents Towards Bioavailable Metal Ions” may be useful for explanation of the results of our earlier studies, concerning the influence of the extract from Aronia melanocarpa L. berries on the body status of zinc and copper.

Reviewer comment

The current form of chapter Discussion is maybe suitable for a food science journal. There is only very limited discussion on the chemical behavior of the studied ligands and complexes. Many of the conclusions are speculations that are not supported by experimental evidence.

Authors’ response

There is only very limited discussion on the chemical behaviour of the studied ligands and complexes because our study was limited to reveal whether the aronia extract and its chosen polyphenolic compounds are capable of binding Zn2+ and Cu2+ at various pH. Our study was not aimed to investigate chemical behaviour of complexes of polyphenolic compounds with zinc and copper and thus the chemical behavior of the studied ligands and complexes cannot be wider discussed to avoid speculations. The discussion section has been corrected in regard of avoiding conclusions being speculations that are not supported by experimental evidence. The authors would like to underline that all conclusions of the study are drawn very carefully. The used terms such as “It may be concluded that … may be explained”, “The findings seem to suggest that…”, “…seem to show…”, “The results may suggest…” show that our conclusions are very careful. We are aware that some of our considerations carried out in the discussion of the results of the present study are in the sphere of supposes that require scientific confirmation; however, this is difficult to avoid when undertaking new research topics and they are important because indicate the possible health implications of bioelements complexation. Currently, there is a great deal of emphasis on seeking practical aspects of results of scientific research, and our work points to the possible health implications of complexing of basic bioelements such as zinc and copper.

Reviewer comment

The concept of chapter Materials and Methods is not suitable. The used procedures should be described in details but without discussion in this chapter. All discussion should be moved to corresponding chapter Discussion.

The presented form is of chapter Materials and Methods difficult to read and follow. The concentrations should be given as mmol/L, amounts of ligands and metals should be given in milligrams and/or milimols. Volumes of all solutions should be given.

Authors’ response

According to the Reviewer suggestion appropriate corrections have been made the Materials and Methods section. Part of the text has been moved into the Discussion section. The Materials and Methods section has been divided into 4 subsections to make it more readable.

Because the primary goal of our study presented in the current manuscript was to investigate whether the 0.1% aronia extract revealed by us to offer protection against various effects of toxic action of cadmium (including disturbances in the body status of zinc and copper), may bind zinc and copper, the concentrations of the extract and particular polyphenols was also expressed as percentage concentration. Because aronia extract contain very numerous ingredients, there is no possibility to express the concentration of extract as mmol/L. That is why, the concentrations of particular polyphenolic compounds, being the ingredient of the extract, as well as the concentration of ethylenediaminetetracetic acid are expressed as percentage concentrations.

Metals concentrations are expressed as mM (M = mol/L, mM = mmol/L).

Volumes of all solutions (aronia extract, particular polyphenolic compounds, solutions of zinc and copper salts) have been provided (Materials and Methods section).

Apart from the above-described changes made in response to the remarks of the Reviewer 4, some corrections have been made in the manuscript in response to the remarks of the Reviewer 1. Because two other Reviewers have no any remarks, all changes made in the manuscript resulted from the remarks of the Reviewer 1 and Reviewer 4.

Once again the authors would like to thank the Reviewer 4 for contribution in improving the manuscript.

If any additional corrections or explanations are needed we are ready to do them immediately.

Yours faithfully,

Malgorzata Brzóska

on behalf of all co-authors

Round 2

Reviewer 1 Report

The efforts shown by the authors to improve the manuscript and the experiments added in the main text and SI, make this manuscript suitable for publication in the present form.

Reviewer 4 Report

I have described significant methodological insufficiencies in my first review. Despite that, authors made only minor corrections of the text. As I have mentioned previously, authors put only minimum effort to describe the studied systems of the model compounds with metal ions. The manuscript is of a very limited scientific value without performing much wider study in terms of pH and metal-to-ligand ratios. Quantification of results and calculation of stability constants are also necessary. Thus, I keep my decision and suggest rejecting the manuscript.

All my remarks from the first review are also kept (except the requirement on deleting the tables of absorption badn maxima from the main text) as almost non of the remarks has been resonably reflected in the revised version of manuscript.